# Spinal cord injury causes chronic bone marrow failure

Randall S. Carpenter [1,2,3,4], Jessica M. Marbourg [1,2,3,4], Faith H. Brennan [2,3,4], Katherine A. Mifflin [2,3,4], Jodie C. E. Hall [2,3,4], Roselyn R. Jiang[2,3,4], Xiaokui M. Mo [5], Malith Karunasiri[6], Matthew H. Burke[6], Adrienne M. Dorrance[6,7] & Phillip G. Popovich [2,3,4 ✉]

Spinal cord injury (SCI) causes immune dysfunction, increasing the risk of infectious morbidity and mortality. Since bone marrow hematopoiesis is essential for proper immune function, we hypothesize that SCI disrupts bone marrow hematopoiesis. Indeed, SCI causes excessive proliferation of bone marrow hematopoietic stem and progenitor cells (HSPC), but these cells cannot leave the bone marrow, even after challenging the host with a potent inflammatory stimulus. Sequestration of HSPCs in bone marrow after SCI is linked to aberrant chemotactic signaling that can be reversed by post-injury injections of Plerixafor (AMD3100), a small molecule inhibitor of CXCR4. Even though Plerixafor liberates HSPCs and mature immune cells from bone marrow, competitive repopulation assays show that the intrinsic long-term functional capacity of HSPCs is still impaired in SCI mice. Together, our data suggest that SCI causes an acquired bone marrow failure syndrome that may contribute to chronic immune dysfunction.

[1] Neuroscience Graduate Program, The Ohio State University, Columbus, OH, USA. [2] Department of Neuroscience, The Ohio State University, Columbus, OH, USA. [3] Belford Center for Spinal Cord Injury, The Ohio State University, Columbus, OH, USA. [4] Center for Brain and Spinal Cord Repair, The Ohio State University, Columbus, OH, USA. [5] Center for Biostatistics and Bioinformatics, The Ohio State University, Columbus, OH, USA. [6] Comprehensive Cancer Center, The Ohio State University, Columbus, OH, USA. [7] Division of Hematology, The Ohio State University, Columbus, OH, USA. ✉email: phillip.popovich@osumc.edu

In adults, mature immune cells develop from a pool of hematopoietic stem and progenitor cells (HSPCs) through a process known as hematopoiesis. Hematopoiesis occurs primarily in bone marrow, where complex cellular interactions and molecular signaling pathways regulate the renewal of millions of immune cells each day[1]. Ultimately, the maintenance of bone marrow hematopoiesis is essential for effective host defense and tissue repair.

Under physiological conditions, the proliferation, differentiation, and retention/release of bone marrow cells, including HSPCs, are controlled by neuroendocrine hormones and the autonomic nervous system, specifically sympathetic neurons[2–12]. After spinal cord injury (SCI), brain and brainstem connections that normally control spinal sympathetic preganglionic neurons are lost, creating a decentralized spinal autonomic network that includes aberrant sympathetic and neuroendocrine reflexes[13]. Uncontrolled sympathetic reflexes have been implicated in the overstimulation and cytotoxicity of mature leukocytes[14–18] and disruption of leukocyte homing after SCI[19].

A decentralized bone marrow may also impair hematopoiesis and contribute to the chronic immune dysfunction that plagues individuals with SCI[20–27]. Indeed, data from two independent studies in which bone marrow aspirates were analyzed from small cohorts of SCI and control subjects indicate that SCI impairs human bone marrow stem cell function[21,22]. Notably, SCI increased the overall proliferation and total numbers of HSPCs in bone marrow of human SCI subjects; however, the ability of these stem cells to form mature immune cells was also impaired[21,22].

The goal of the current study is to determine the extent of hematopoietic dysfunction after acute and chronic SCI, and to identify molecular, cellular, and physiological mechanisms that may explain any SCI-induced impairments that develop in the bone marrow. We hypothesize that SCI will induce acute and chronic bone marrow failure in mice, similar to what has been described after SCI in humans. Our data show that SCI causes a rapid and chronic bone marrow failure syndrome characterized by excessive HSPC proliferation, accumulation, and impaired function. Importantly, post-injury injections of Plerixafor, an FDA-approved drug that blocks CXCR4, liberates HSPCs from bone marrow and partly reverses bone marrow failure by promoting extramedullary hematopoiesis. Treating bone marrow failure after SCI may help to reverse chronic immune dysfunction and anemia that persist indefinitely after SCI in humans.

## Results

**SCI increases bone marrow cell proliferation.** Hematopoiesis, or the formation of new red and white blood cells, requires that HSPCs proliferate and differentiate in the marrow of all bones, including long bones (i.e. femur/tibia) and sternum. To investigate how SCI generally affects cell proliferation in bone marrow, we monitored post-injury changes in the sternum and femur/tibia of transgenic female mice that express luciferase in dividing cells (Mito-luc mice[4,28,29]; Fig. 1a). We used peak (max radiance) and average (total flux) bioluminescence as representative measures of bone marrow proliferation. To control for potential effects of surgical stress on cell proliferation, SCI mice were compared with sham-injured mice receiving a laminectomy only (Lam). Within 72 h of surgery, cellular proliferation increased in the sternum and femur of sham-injury and SCI mice; however, enhanced proliferation was maintained only in SCI mice, peaking within the first 7 days post-injury (dpi) and persisting up to 1-month post-injury (latest time evaluated) (Fig. 1b, c).

To determine whether these changes could be specifically attributed to enhanced proliferation of HSPCs, multi-color flow cytometry was used to quantify the proportion of proliferating

lineage⁻, Sca-1⁺, c-Kit⁺ (LSK) cells in the femoral bone marrow of C57BL/6 wild-type mice after SCI (Supplementary Fig. 1a, b)[30]. At 3 dpi, LSK cell proliferation increased significantly (Fig. 1d). In separate cohorts of mice, the mitogenic effect of SCI on LSK cells was confirmed and was found to be long lasting (Fig. 1e); total numbers of differentiated c-Kit⁺ progenitors (Fig. 1f) and mature bone marrow cells increased until at least 28 dpi (Fig. 1g, h). Together, these data indicate a reactive bone marrow response after SCI marked by protracted proliferation and accumulation of HSPCs.

**SCI causes chronic expansion of bone marrow HSCs.** The LSK fraction of bone marrow cells characterized in Fig. 1 includes all stem and multipotent progenitor cells[31,32]. To better define how SCI affects this pool of phenotypically and functionally heterogeneous cells, multi-color flow cytometry was used to analyze cells from another group of sham-injured and SCI mice (Fig. 2a, Supplementary Fig. 1c). Consistent with data in Fig. 1, the total number of bone marrow cells (Fig. 2b), including total LSK cells (Fig. 2c), increased after SCI. The increase in cell number affects multiple subsets of stem and progenitor cells including long-term repopulating stem cells (LT-HSCs; CD150⁺/CD48⁻/CD135⁻ LSKs), short-term HSCs (ST-HSCs; CD150⁻/CD48⁻/CD135⁻ LSKs), multipotent progenitor subset 2 cells (MPP2s; CD150⁺/CD48⁺/CD135⁻LSKs), myeloid-primed MPP3s (CD150⁻/CD48⁺/CD135⁻LSKs), lymphoid-primed MPP4s (CD135⁺ LSKs), and granulocyte–monocyte progenitors (GMPs; CD16/32⁺ LK cells) (Fig. 2d–i). Importantly, SCI did not cause HSPCs to increase expression of γH2AX, a marker of double-stranded DNA breaks (Fig. 2j). These data demonstrate that SCI causes excessive proliferation of all HSPCs but without evidence of long-term replication stress.

**SCI prevents HSPC mobilization from bone marrow.** Enhanced HSPC proliferation after sham injury or SCI may represent a compensatory response to the stress of blood loss and trauma. Indeed, both psychological and physical stressors enhance bone marrow hematopoiesis and extramedullary hematopoiesis in secondary lymphoid tissues, including the spleen[4,33–37].

We confirmed in the present study an acute effect of surgical stress on bone marrow hematopoiesis. Three days after sham surgery, extensive numbers of HSPCs were found in blood and spleen of sham-injured mice (occurring above circulating numbers of HSPCs in naïve mice) as assessed by MethoCult (Fig. 3a, b) and flow cytometry (Fig. 3d, e) assays, leading to an increase in spleen size (Fig. 3c). In contrast, 3 days after SCI, fewer HSPCs entered the circulation (Fig. 3a) or colonized the spleen in SCI mice (Fig. 3b, d, e). Additionally, fewer c-Kit⁺ HSPCs were proliferating in the spleen of SCI mice (Fig. 3e), likely as fewer proliferating HSPCs exit the bone marrow and enter the circulation after SCI.

The effects of SCI on HSPC sequestration in bone marrow were independent of sex; SCI abolished HSPC mobilization in both female and male SCI mice (compare Fig. 3a–c and g–i). Further, SCI-dependent effects on HSPCs were not influenced by either injury severity or spinal injury level; HSPCs fail to enter into the circulation after complete spinal transection injuries at either L6/S1, T9, or T3 spinal levels (Fig. 4a) and after an incomplete spinal contusion injury at either T3 or T9 spinal levels (Fig. 4b).

To determine if the SCI-dependent effects on HSPCs were species-specific, male and female mice with human HSPCs and immune systems (i.e., humanized mice) were generated[38–40]. Similar to wild-type mice, SCI prevented human HSPCs from entering the circulation and trafficking to the spleen (Fig. 4c, d).

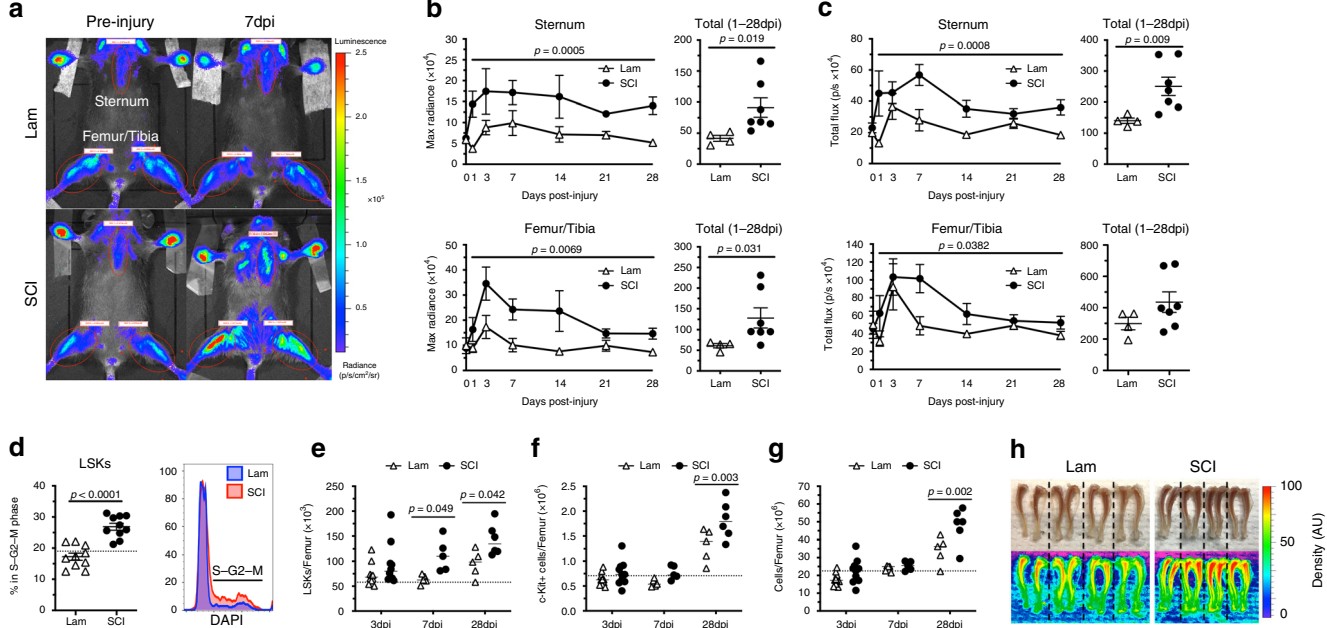

**Fig. 1 T3 transection SCI protracts HSPC proliferation and causes HSPC accumulation in bone marrow. a** Bioluminescent imaging of sternum and femur/tibia regions of interest (ROIs; red ellipses) pre-injury and 7 dpi. **b, c** Bioluminescence expressed as max radiance (peak intensity within ROI) and total flux (average intensity within ROI) for each region, with left and right femur/tibia averaged; data also expressed as the total (sum) of signals from 1 to 28 dpi. **d** Proportion of proliferating lineage low, Sca-1+, c-Kit+ (LSK) bone marrow cells, as indicated by the S–G2–M phase of the cell cycle, at 3 dpi; representative histograms of LSK proliferation at 3 dpi (closest to the mean). Quantification of LSKs (**e**), c-Kit+ progenitors (**f**), and total bone marrow cells (**g**) at 3, 7, and 28 dpi. **h** Images and heatmaps of tibia pairs from Lam and SCI mice demonstrating increased optical density (artificial units) corresponding with increased bone marrow cellularity. All data are mean ± SEM, mixed effect model with repeated measures (**b, c**), two-sample *t*-test (**b–d**), two-way ANOVA with Bonferroni multiple comparisons (**e–g**); *n* = 4 and 7/group in **b** and **c**, *n* = 10/group in **d**, *n* = 10/group (3 dpi), 5/group (7 dpi), and 5 and 6/group (28 dpi) in **e–g**. Dotted lines represent average data from naïve mice. dpi days post-injury, Lam laminectomy (sham injury). Source data are provided as a Source Data file.

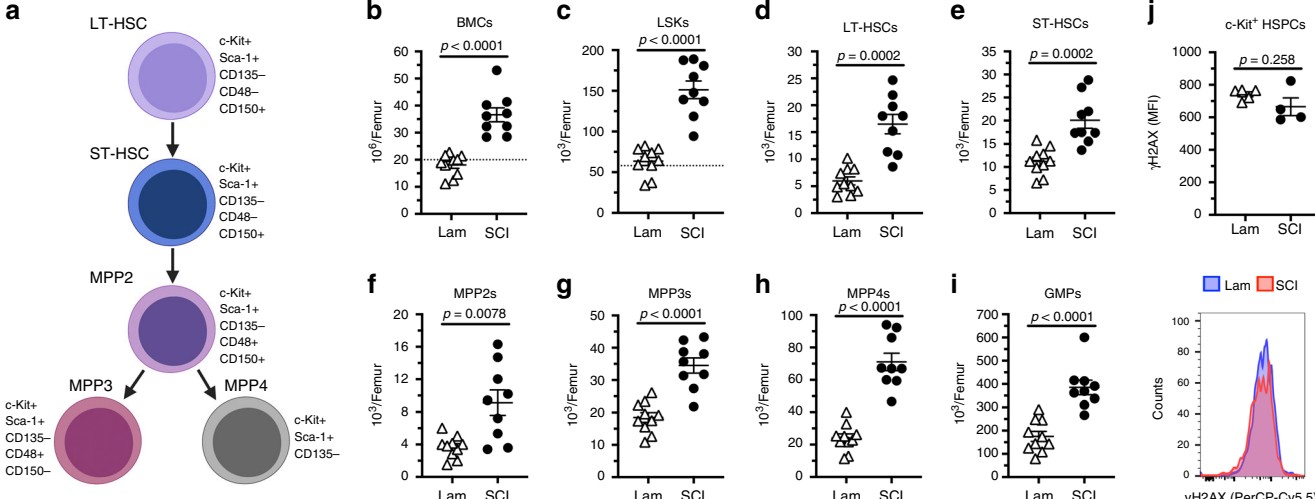

**Fig. 2 T3 transection SCI causes the chronic accumulation of hematopoietic stem cells and differentiated progenitors. a** Hematopoietic hierarchy of HSPCs and their cell surface markers. Total number of bone marrow cells (**b**), LSKs (**c**), long-term HSCs (**d**), short-term HSCs (**e**), multipotent progenitors 2–4 (**f–h**), and granulocyte–monocyte progenitors (**i**) at 28 dpi. **j** Mean fluorescence intensity of γH2AX expression (i.e. replication stress) within c-Kit+ HSPCs at 28 dpi. All data are mean ± SEM, two-sample *t*-test; *n* = 9 and 10/group (**b–i**), *n* = 5 and 4/group (**j**). Dotted lines represent average data from naïve mice. dpi days post-injury, Lam laminectomy (sham injury). Source data are provided as a Source Data file.

Together, these data, when combined with data in Figs. 1–3, indicate that SCI triggers excessive proliferation of HSPCs but that these cells are unable to leave the bone marrow niche, causing HSPCs to progressively accumulate within the bone marrow. Importantly, HSPC sequestration in bone marrow, although SCI-dependent, is independent of sex, injury level, injury severity, or species (both mouse and human HSPCs respond identically to SCI).

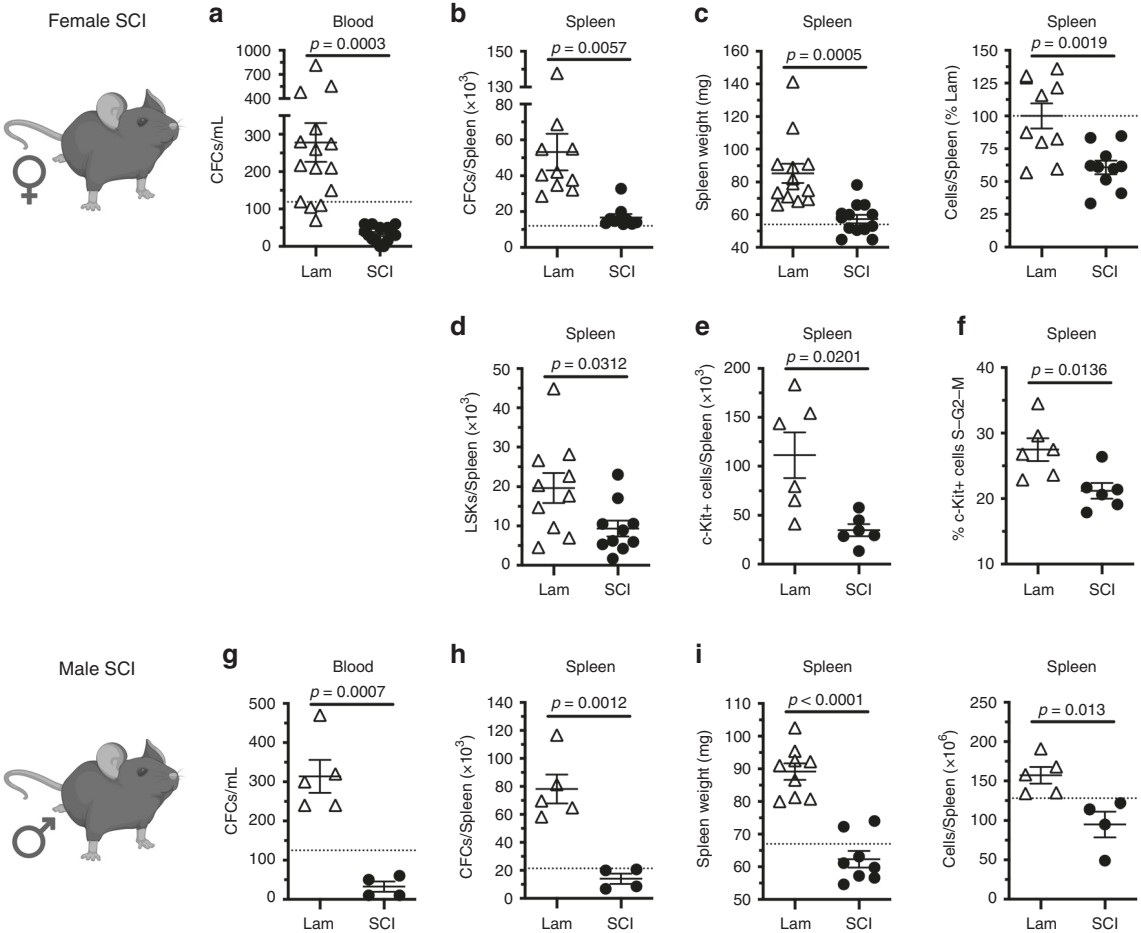

**Fig. 3 T3 transection SCI prevents acute HSPC mobilization into blood and trafficking to the spleen at 3 dpi. a–f** Outcomes in female mice at 3 dpi. **a** Total numbers of CFCs per mL of whole blood. **b** Total numbers of CFCs per spleen. **c** Spleen weight and number splenocytes (% of Lam). **d**, **e** Total numbers of (**d**) LSK and (**e**) c-Kit$^+$ cells in the spleen. **f** Proportion of proliferating c-Kit$^+$ cells (S–G2–M phase) in the spleen. **g–i** Identical parameters as measured in **a–c** in male mice at 3 dpi. All data are mean ± SEM, two-sample $t$-test; $n = 15$/group in **a**; $n = 10$/group in **b–d** (15/group spleen weight); $n = 6$/group in **e**, **f**; $n = 4$ and 5/group **g–i** (8 and 9/group spleen weight). Dotted lines represent average data from naïve mice. dpi days post-injury, Lam laminectomy (sham injury), CFCs colony forming cells. Source data are provided as a Source Data file.

**SCI impairs B cell development and mobilization.** The decentralized bone marrow that develops after SCI may sequester cells other than HSPCs. Indeed, in both animal models and human subjects, total numbers of circulating leukocytes decrease after SCI and stroke[19,24,41–44]. Here, we confirmed and expanded those data using a model of complete SCI (T3 spinal level). Specifically, at 3 dpi, T3 transection SCI reduced circulating blood lymphocytes (Fig. 5a, b) with a concomitant increase in the proportion of mature lymphocytes found in bone marrow, including B220$^{hi}$ B cells, CD3$^+$/CD4$^-$ T cells, and CD3$^+$/CD4$^+$ T cells (Fig. 5c–e). Further, SCI reduced numbers of B220$^{low}$ immature B cells (confirmed to be enriched for ProB, PreB, and IgM$^{low}$ immature B cells; Supplementary Fig. 1e) in the bone marrow (Fig. 5c), similar to previous data showing impaired B lymphopoiesis after SCI and stroke[19,44,45].

**SCI increases CXCL12-CXCR4 signaling in bone marrow HSPCs.** Immunoregulatory cytokines and chemokines, including IL1β, TNFα, TGFβ, CCL2, CXCL12, and CXCR4 can affect the development and function of HSPCs[46,47]. To determine if the excessive proliferation and retention of HSPCs in bone marrow after SCI is associated with changes in these cytokines, we prepared mRNA from whole-bone marrow cells isolated from T3

transection SCI mice and sham-injured mice. Real-time PCR analyses revealed that, with the exception of Ccl2, SCI increased the expression of all bone marrow cytokines and chemokines assessed (Fig. 6a). Notably, Cxcl12, a chemokine produced by bone marrow stromal cells, and its receptor Cxcr4 were increased in parallel (fourfold and threefold, respectively).

Several published studies demonstrated that CXCL12 binding to CXCR4 on HSPCs influences the maintenance and retention of HSPCs in bone marrow, a phenomenon that is dependent, in part, on the sympathetic branch of the autonomic nervous system[8,10,48]. Therefore, we validated the effects of high-level complete (T3 transection) SCI on this chemokine/chemokine receptor pair. In a separate cohort of T3 SCI and sham-injured mice, we found that SCI increased the amount of secreted CXCL12 protein in bone marrow extracellular fluid (Fig. 6b) and expression of CXCR4 receptor on LSK cells (Fig. 6c).

To test whether the SCI-dependent increase in CXCL12/CXCR4 expression causes HSPC sequestration in bone marrow, mice were treated with Plerixafor (AMD3100), a small-molecule antagonist of CXCR4, or vehicle for 3 days, beginning 1 h post-SCI. In Plerixafor-treated mice, the number of HSPCs released into the blood increased >6-fold (Fig. 6d), with a corresponding increase in the number of HSPCs found in the spleen (Fig. 6e–g).

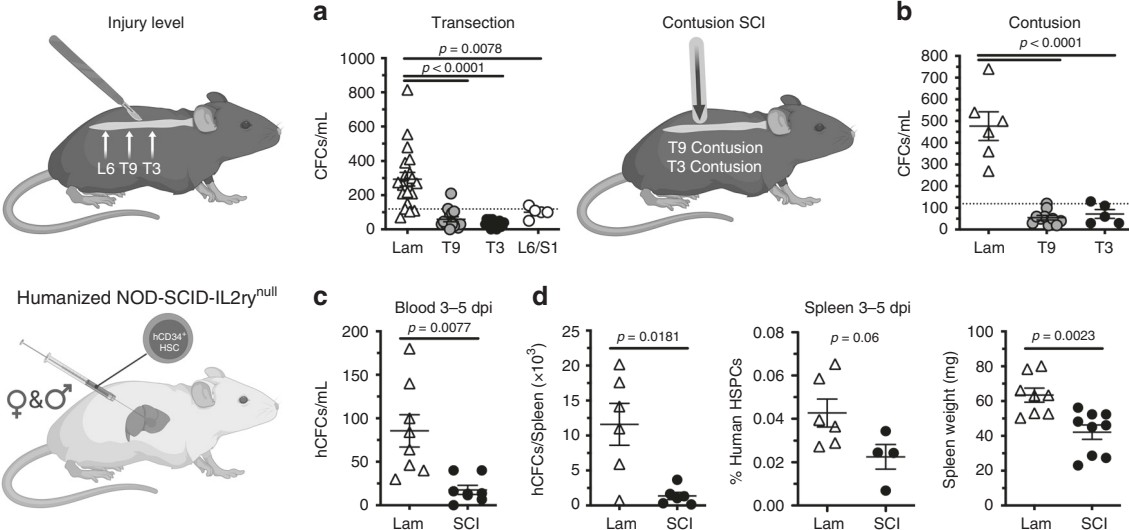

**Fig. 4 Complete and incomplete SCI impairs acute mouse and human HSPC mobilization. a** Number of CFCs per mL whole blood after T3, T9, or L6/S1 transection injury. **b** Number of CFCs per mL whole blood after T3 and T9 contusion injury. **c** Number of hCFCs per mL whole blood. **d** Total numbers of splenic hCFCs, proportion of splenic hCD34hi/hCD38low human HSPCs, and spleen weight of hNSG mice 3–5 dpi. All data are mean ± SEM, one-way ANOVA with Tukey multiple comparison test (**a**, **b**), two-sample $t$-test (**c**, **d**); $n = 20, 15, 14$, and 5/group in **a**; $n = 6, 10$, and 5/group in **b**; $n = 8$/group in **c**; $n = 6, 6; 6, 4; 8$ and 9/group in **d**. dpi days post-injury, Lam laminectomy (sham injury). Source data are provided as a Source Data file.

Plerixafor treatment also reversed the normal onset of post-SCI leukopenia; both total numbers of splenocytes (Fig. 6h) and all subsets of circulating WBCs increased (Fig. 6i, j). Importantly, releasing cells from bone marrow with Plerixafor did not ablate the bone marrow. In fact, Plerixafor increased total bone marrow cellularity (Fig. 6k). These data implicate sustained CXCL12/CXCR4 signaling within the bone marrow as a potential mechanism underlying SCI-induced sequestration of HSPCs. Importantly, post-SCI sequestration of HSPCs and mature immune cells can be overcome using the FDA-approved drug, Plerixafor.

**Chronic SCI impairs bone marrow responses to inflammatory stimuli.** Data above indicate that SCI causes an acquired bone marrow failure syndrome characterized by protracted HSPC proliferation with sequestration and associated peripheral lymphopenia. Bone marrow failure develops within 72 h post-injury and persists until at least 1-month post-injury. To determine whether failed bone marrow in SCI mice is able to respond to physiologically relevant stimuli, mice with chronic SCI were challenged with endotoxin (1 mg kg⁻¹; lipopolysaccharide (LPS)), a potent inducer of HSPC proliferation and mobilization[34,49–51]. Mito-luc transgenic and C57BL/6J (wild-type) mice were injected with LPS daily for 3 days beginning 6 weeks after T3 transection SCI or sham surgery (Fig. 7a).

In all sham and SCI Mito-luc transgenic mice, cellular proliferation in bone marrow decreased 2–4 days after the first LPS injection (Fig. 7c). This effect was expected since LPS stimulates the egress of proliferating cells from the bone marrow into the circulation[34,52]. Indeed, when compared to total number of HSPCs found in the blood of non-LPS naïve mice, LPS significantly increased total HSPCs in blood in all mice. However, fewer HSPCs were detected in the blood of SCI mice at both peak mobilization (4 days after first dose of LPS) and during recovery (11 days after first dose of LPS) (Fig. 7d). Strikingly, LPS stimulation reduced the proportion and number of circulating lymphocytes in SCI mice, but not in sham-injured mice (Fig. 7e). By 7 days after LPS injections, basal cell proliferation is restored and maintained in sham-injured mice, presumably

because HSPCs have repopulated the depleted bone marrow[49]. In contrast, LPS-induced proliferation continues in the bone marrow of SCI mice; the luciferase signal overshoots baseline proliferation by >200% between 9 and 14 days post-LPS (Fig. 7b, c). Remarkably, the same characteristics that define bone marrow failure early after SCI are recapitulated in chronic SCI animals after LPS injections (compare Fig. 7 to Figs. 1–5). Collectively, these data indicate that SCI causes chronic, and perhaps permanent, bone marrow failure.

**SCI impairs the long-term function of bone marrow HSPCs.** Data in Fig. 6 indicate that the sequestration and accumulation of HSPCs in bone marrow after SCI are due, in part, to aberrant cytokine and chemokine signaling in the bone marrow niche. To determine whether SCI also affects the intrinsic capacity of bone marrow HSPCs, notably their ability to restore hematopoiesis to a depleted bone marrow niche, we performed in vivo competitive repopulating assays.

Bone marrow HSPCs were removed from SCI mice 3 days after T3 transection SCI or sham surgery. In SCI mice, this timing corresponds with a period of enhanced HSPC proliferation (Fig. 1) and sequestration (Fig. 3), but no difference in total HSPC numbers (Fig. 1). When bone marrow cells were injected into uninjured mice that had received lethal doses of irradiation (Supplementary Fig. 2a, b), SCI-derived donor cells exhibited faster engraftment in recipient mice at 8 weeks (Fig. 8a, b). By 19 weeks post-engraftment, stable engraftment was achieved in all recipient mice, regardless of the donor cell source (Fig. 8b). Notably, the enhanced engraftment potential of SCI bone marrow cells relative to sham donor cells waned after 8 weeks (arrowhead, Fig. 8a, c). However, we did not observe long-term lineage biasing of engrafted SCI donor cells through 19 weeks (Supplementary Fig. 2e). These data indicate a possible deficit in the self-renewal capacity, but not differentiation, of long-term hematopoietic stem cells from SCI donors (Fig. 8c, d).

To test whether SCI impairs the self-renewal capacity of long-term HSPCs, donor bone marrow was isolated from primary recipients after 19 weeks and then re-transplanted into new lethally irradiated recipient mice (Supplementary Fig. 2a).

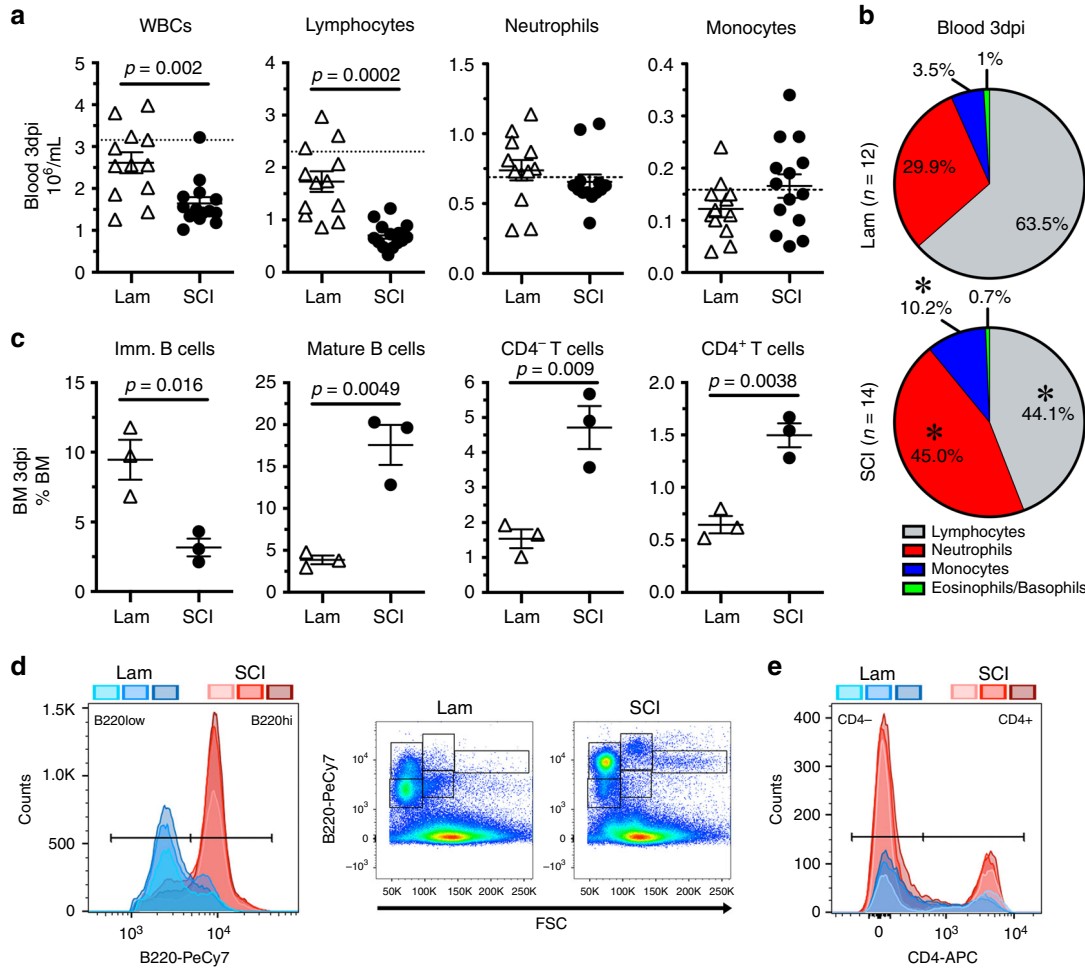

**Fig. 5 T3 transection SCI causes acute lymphopenia in blood with concurrent sequestration of lymphocytes in bone marrow. a** Total numbers of white blood cells (WBCs), lymphocytes, neutrophils, and monocytes per mL of whole blood at 3 dpi. **b** Lymphocytes (gray), neutrophils (red), monocytes (blue), and eosinophils/basophils (yellow) as a proportion of total WBCs. **c** Proportion of immature (i.e. developing) B cells (B220$^{low}$), mature B cells (B220$^{hi}$), and mature CD3$^+$ T cells (CD4$^-$ and CD4$^+$ subsets) in bone marrow. **d** Flow cytometry gating of bone marrow B220$^+$ B cells demonstrating reduced numbers of immature (B220$^{low}$) and increase numbers of mature (B220$^{hi}$) B cells with SCI. Examples of B220 and forward scatter (size) characteristics of B cells in bone marrow after sham injury and SCI. **e** Flow cytometry gating of bone marrow CD3$^+$/CD4$^-$ and CD3$^+$/CD4$^+$ mature T cell subsets demonstrating increased numbers after SCI. All data are mean ± SEM, *p < 0.05, two-sample t-test; n = 12 and 14/group in **a**, **b**; n = 3/group in **c**–**e**. Shades of blue/red (Lam/SCI) represent individual sample plots in **d** and **e**. Dotted lines represent average data from naïve mice. dpi days post-injury, Lam laminectomy (sham injury), FSC forward scatter, BM bone marrow. Source data are provided as a Source Data file.

HSPC engraftment was identical between groups 4 weeks after secondary transplantation (Fig. 8e, f). However, no significant change in engraftment occurred throughout the evaluation period (6 months) in mice receiving SCI bone marrow; only mice receiving HSPCs that were originally derived from sham-injured mice demonstrated increases in donor engraftment (Fig. 8e–g). Importantly, SCI donor cells produced significantly fewer donor WBCs than sham-injured donor cells (Supplementary Fig. 2h). By 24 weeks after secondary transplantation, engraftment was significantly reduced in the bone marrow of mice receiving SCI donor cells compared with mice receiving sham-injured donor cells (Fig. 8h). These data indicate that SCI negatively affects the long-term self-renewal capacity of bone marrow HSPCs.

## Discussion

Prior studies in mice[45] and humans[22] found that SCI increases the total number of cells in bone marrow. Data in this report extend those observations and also provide new insight to help explain how/why after SCI, both humans[24,41,53] and rodents[42,43,54,55] exhibit prolonged hematological abnormalities marked by leukopenia and chronic immunologic dysfunction[18,23,26,56]. Specifically, data in this report reveal that traumatic SCI causes a bone marrow failure syndrome marked by excessive proliferation and sequestration of HSPCs, altering of cytokine and chemokine signaling within the bone marrow niche, failure to generate and mobilize mature lymphocytes, and chronic impairments in the clonogenic potential of HSPCs (Fig. 9). Bone marrow failure diseases develop when the bone marrow is unable to produce appropriate numbers of healthy mature white and red blood cells. Normal aging[12,57] and various diseases including diabetes[58,59], warts, hypogammaglobulinemia, infections, and myelokathexis (WHIM) syndrome[60–62], glioblastoma[63], and chemotherapy[64] cause sequestration of mature and immature cells in the bone marrow. In each case, sequestration of bone marrow cells either causes or is associated with hematopoietic dysfunction. Both neural and humoral mechanisms undoubtedly participate in causing the aberrant sequestration phenomenon, although disease-specific mechanisms are likely.

HSPCs reside in a specialized perivascular bone marrow niche (i.e. microenvironment)[1]. It is in the niche that complex cellular

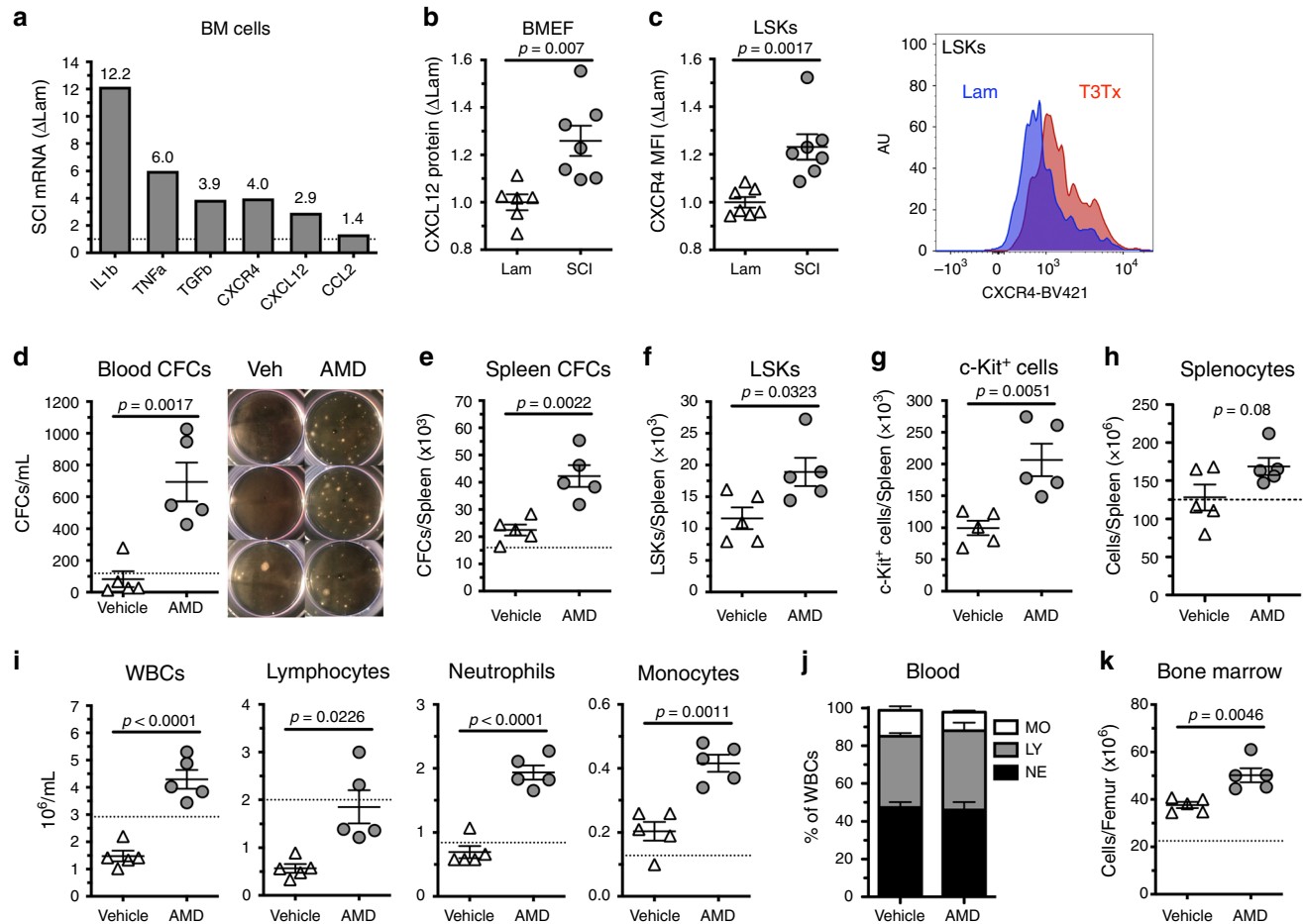

**Fig. 6 Plerixafor (AMD3100) mobilizes HSPCs, rescues extramedullary hematopoiesis, and boosts circulating immune cells after T3 transection SCI. a** Cytokine and chemokine mRNA expression from whole-bone marrow cells of SCI mice 3 dpi (expressed as fold change of Lam). **b** CXCL12 protein expression from bone marrow extracellular fluid extracts. **c** Protein expression (MFI) of CXCR4 expression on LSKs from whole bone marrow. **d** Total CFCs per mL whole blood, **e** CFCs per spleen, **f** LSKs per spleen, **g** c-Kit+ HSPCs per spleen, and **h** total cells per spleen 3 days after SCI with and without daily AMD3100 treatment (once per day). **i** Total white blood cells (WBCs), lymphocytes, neutrophils, and monocytes per mL whole blood. **j** Proportion of monocytes (MO), lymphocytes (LY), and neutrophils (NE) in whole blood ($n = 5$/group). **k** Total cells per femur. All data are mean ± SEM, two-sample $t$-test; $n = 3$/group (pooled) in **a**; $n = 6$ and 7/group in **b**; $n = 7$/group in **c**; $n = 5$/group in **d–k**. Dotted lines represent average data from lam (**a**) or naïve mice (**d, e, h, i, k**). BM bone marrow, MFI mean fluorescence intensity, WBCs white blood cells, MO monocytes, LY lymphocytes, NE neutrophils, AMD Plerixafor (AMD3100). Source data are provided as a Source Data file.

and molecular cues orchestrated between HSPCs, CXCL12-expressing perivascular stromal cells and the autonomic nervous system, particularly sympathetic noradrenergic nerve fibers, control HSPC proliferation, regeneration, and differentiation[8,10,11,65]. Specifically, post-ganglionic sympathetic neurons release norepinephrine into the bone marrow, causing bone marrow stromal cells to decrease their expression of CXCL12 which, in turn, untethers CXCR4+ HSPCs and mature leukocytes from the niche, facilitating their egress into the circulation[8,10,36]. Loss of proper sympathetic tone in bone marrow, whether by experimental manipulation or by aging, causes hematopoietic dysfunction[12,64]. A similar neurogenic mechanism may cause bone marrow failure after SCI.

After SCI, much or all of the tonic supraspinal control over the sympathetic nervous system is lost. Previously, using models of thoracic SCI, we showed that normal physiological stimuli (e.g., visceral afferent input from bladder/bowel) trigger exaggerated or uncontrolled sympathetic reflexes in spinal autonomic circuits[14–17]. However, similarly aberrant sympathetic reflexes may be triggered even when SCI occurs below the caudal-most sympathetic preganglionic neurons located in lower thoracic/upper lumbar spinal cord. Anatomical tracing studies indicate that femoral bone marrow is innervated by both sympathetic and sensory nerves originating as far rostral in the spinal cord as T8-9 and as far caudal as the sacral spinal cord[7,66,67]. This is a large segment of spinal cord through which sensory input from bone marrow could activate propriospinal relay neurons and multi-segmental sympathetic reflexes[13]. Depending on which relay neurons are activated and the relative integrity (anatomical or functional) of the intersegmental circuitry, bone-specific intersegmental sympathetic reflexes may be exaggerated or silenced after SCI. At the level of the bone marrow niche, either outcome would be perceived by stromal cells and HSPCs as a break in homeostasis. Under pathological conditions, when sympathetic tone to bone marrow is disrupted, CXCL12 and CXCR4 levels increase and HSPC mobilization is impaired[10,12,59,64]. Since proper descending modulation of spinal sympathetic reflexes is never restored after traumatic SCI and also intersegmental neuronal circuitry continues to undergo structural and functional plasticity, neurogenic control of bone marrow function may never be restored after SCI. Evidence for permanent bone marrow failure is apparent from data in Fig. 7, which

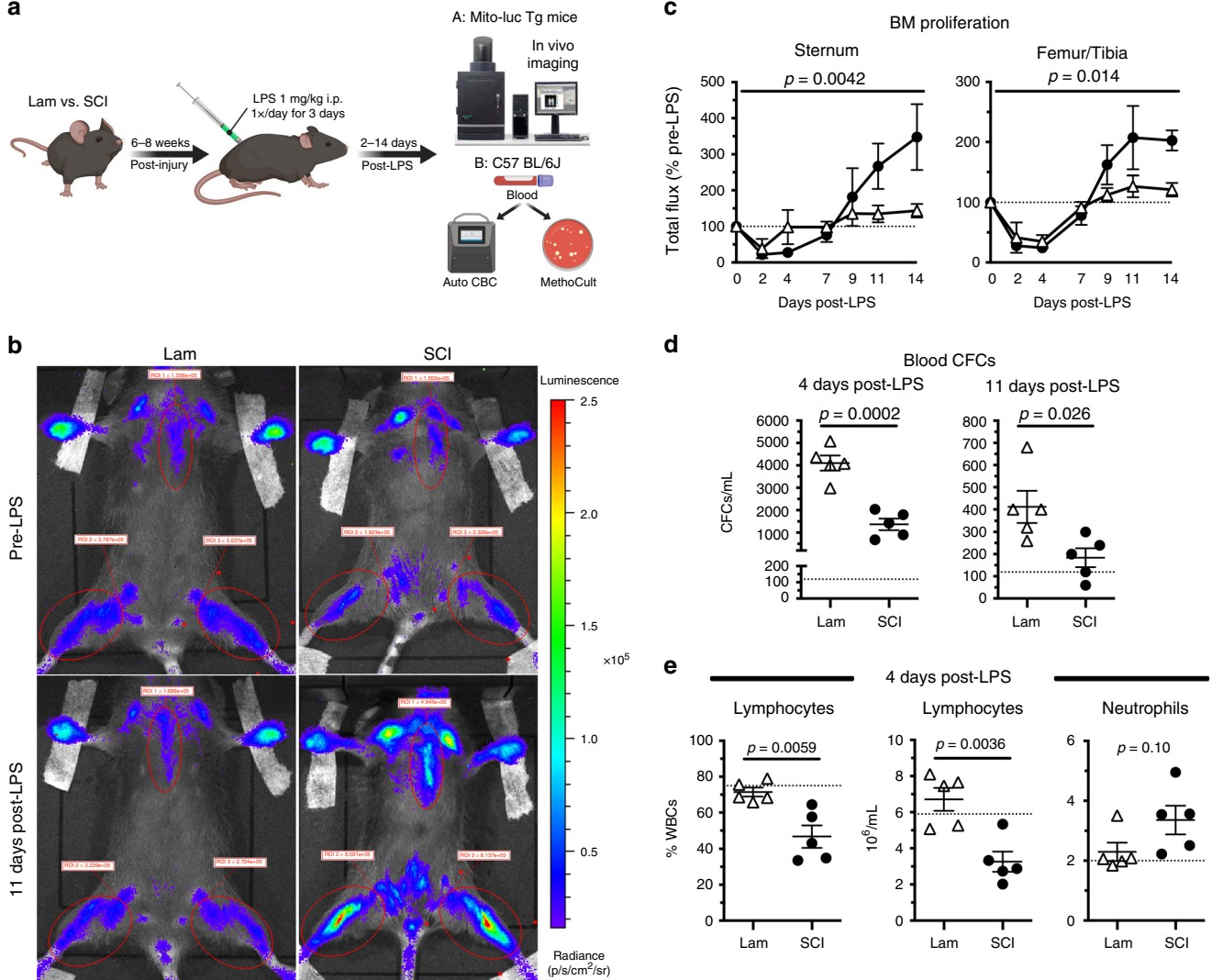

**Fig. 7 Stimulating hematopoiesis with LPS in chronic-injured mice recapitulates acute bone marrow responses to injury. a** Experimental outline demonstrating T3 transection SCI and sham injury followed by 6–8 weeks of recovery prior to LPS stimulation (1 mg kg$^{-1}$ day$^{-1}$ i.p. for 3 days). Mice then underwent in vivo imaging of bone marrow proliferation (A: Mito-luc transgenic mice) or automated CBC and MethoCult assay of whole blood (B: C57BL/6J mice). **b** Bioluminescent images from representative Lam and SCI mice before and 11 days post-LPS. **c** Quantification of total flux in sternum and femur/tibia regions as a % of pre-LPS levels (dotted line). **d** Total number of CFCs per mL whole blood at 4 and 11 days post-LPS. **e** Lymphocytes (% of WBCs and total numbers) and neutrophils (total numbers) in blood 4 days post-LPS. All data are mean ± SEM, two-way ANOVA with repeated measures (**c**) and two-sample t-test (**d**, **e**); n = 4/group in **c**, n = 5/group in **d**, **e**. Dotted lines represent average data from naïve mice (**d**, **e**); days post-LPS refers to time after first LPS dose. Lam laminectomy (sham injury). Source data are provided as a Source Data file.

revealed that a potent hematopoietic stimuli (endotoxin) does not effectively mobilize HSPCs and mature lymphocytes from the bone marrow of mice at chronic post-injury periods.

SCI-induced changes in circulating hormones, glucocorticoids in particular, are also likely culprits underlying acute and chronic bone marrow failure. Normally, circulating glucocorticoids act on diverse cell types in the periphery and the nervous system to help maintain HSPCs and the bone marrow niche[68]. Glucocorticoids also promote HSPC homing from the bone marrow, an effect that is mediated by glucocorticoid receptor-mediated induction of CXCR4 transcription in HSPCs[69]. The acute physical stress of spinal trauma and the sustained activation of aberrant sympathetic–neuroendocrine reflexes cause a primary hypercortisolism after SCI[14–16,19]. As a result, circulating levels of glucocorticoids increase within 24 h post-injury then remain at supraphysiological concentrations indefinitely after SCI. The development of aberrant sympathetic–neuroendocrine reflexes

may explain why, in the present study, SCI always causes HSPC sequestration, regardless of injury level (see Fig. 4). Future studies are needed to explore in-depth how SCI affects neural–humoral signaling in bone marrow and to what extent glucocorticoids and other blood-borne factors (e.g., microbial metabolites) contribute to acquired bone marrow failure.

An important observation made in this report, and one that may have an immediate impact on people affected by SCI, is that it is possible to overcome certain aspects of SCI-induced bone marrow failure. Specifically, we found that the FDA-approved drug and CXCR4 antagonist Plerixafor, when injected post-injury, effectively liberates HSPCs and mature leukocytes from the bone marrow of SCI mice (Fig. 6). In SCI patients, Plerixafor could be a potentially safe and effective way to mobilize cells from the bone marrow niche to help restore immune function[50]. Indeed, Plerixafor can safely reverse immunodeficiency in WHIM patients[62]. Still, to minimize the risk associated with boosting the

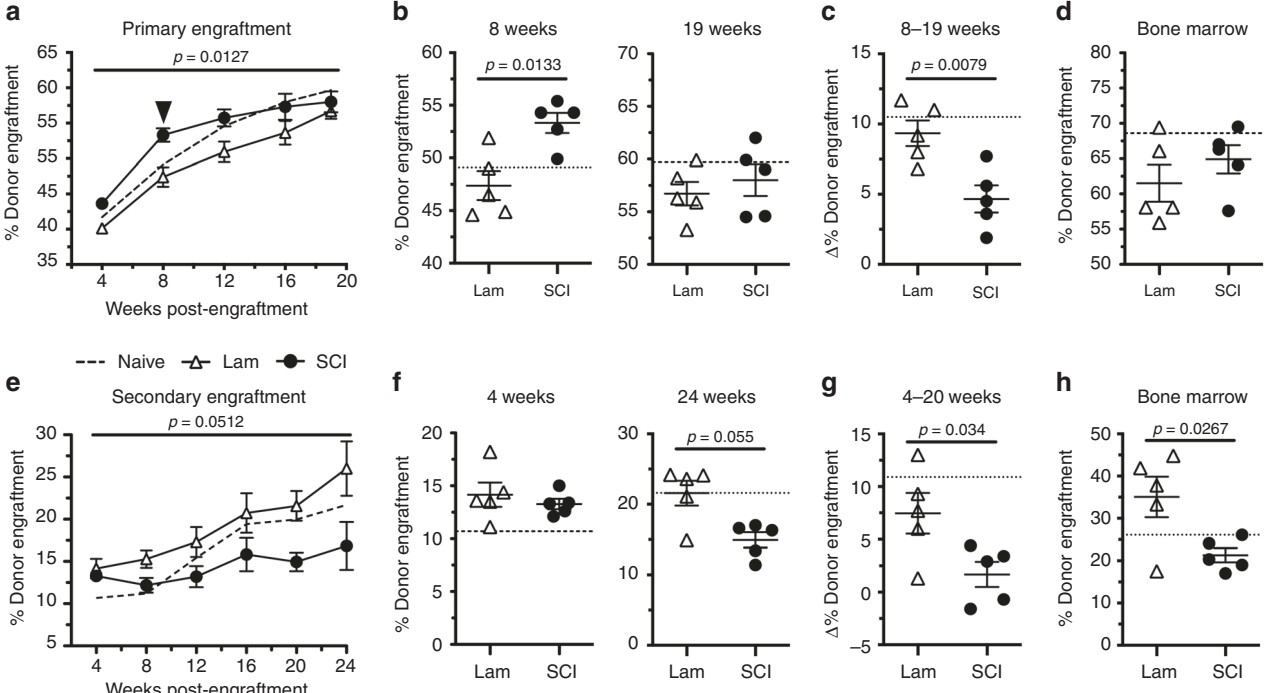

**Fig. 8 T3 transection SCI impairs the long-term clonogenic potential of bone marrow by 3 dpi. a** Percent donor engraftment in recipient BoyJ mice 4–19 weeks after primary transplantation with 3 dpi bone marrow from naïve, sham-injured (Lam), and T3 transection SCI mice. **b** Percent donor engraftment after primary CRU assay at the 8- and 19-week timepoints from **a**. **c** Change in percent donor engraftment from 8 to 19 weeks post-transplantation. **d** Percent donor engraftment in bone marrow of recipient BoyJ mice 19 weeks after primary transplantation. **e** Percent donor engraftment in recipient BoyJ mice 4–20 weeks after secondary transplantation with bone marrow from primary CRU mice. **f** Percent donor engraftment after primary CRU assay at the 4- and 20-week timepoints from **d**. **g** Change in percent donor engraftment from 4 to 20 weeks post-transplantation. **h** Percent donor engraftment in bone marrow of recipient BoyJ mice 25 weeks after secondary transplantation. All data are mean ± SEM, two-way ANOVA with repeated measures followed by Bonferroni multiple comparisons (**a, b, e, f**), two-sample $t$-test (**c, d, g, h**); $n = 5$/group in **a–h**; dotted lines represent averaged data from naïve mice. Source data are provided as a Source Data file.

immune system and exacerbating neuroinflammatory-mediated injury or trauma-induced autoimmunity, both of which can impair neurological recovery, more research is needed to define the optimal therapeutic conditions for Plerixafor treatment after SCI.

Another intriguing aspect of SCI-induced bone marrow failure was revealed during the competitive repopulating unit (CRU) assays (Fig. 8). Specifically, we found evidence that the long-term clonogenic potential of HSPCs is impaired and that these long-term effects are imprinted in HSPCs early after injury. Indeed, bone marrow cells isolated from SCI donor mice 3-day post-injury, when transplanted into irradiated naïve mice, engrafted the irradiated bone marrow faster than bone marrow cells obtained from sham-injured mice. This early advantage could be explained by improved homing to the bone marrow by HSPCs from SCI donors, perhaps because of glucocorticoid-mediated enhancement of CXCR4 on bone marrow LSK cells (Fig. 6). However, this repopulation advantage was transient as deficits in the long-term repopulation or clonogenic potential of SCI bone marrow became obvious after secondary transplantation (Fig. 8). Currently, we do not know why SCI impairs the long-term function of HSPCs or how this occurs within only 3 days. Since we did not see an increase in the DNA damage marker γH2AX after chronic SCI (Fig. 2), it is unlikely that excessive HSPC proliferation after SCI causes DNA damage[70]. However, HSPC exhaustion is possible as cell cycle number has been shown to inversely correlate with long-term HSPC function[71,72]. It is also possible that aberrant sympathetic–neuroendocrine reflexes influence the clonogenic potential of HSPCs; both catecholamines

and glucocorticoids can cause epigenetic modifications in cells, imprinting then with new functional identities[69,73]. In the context of monocytes/macrophages, for example, exposure of these cells to certain stimuli endows them with enhanced microbicidal functions against secondary infections[74]. This trained immunity in monocytes/macrophages is orchestrated by epigenetic reprogramming, which also occurs in HSPCs[75–77]. Perhaps SCI creates an environment in the bone marrow niche that favors the induction of a form negative trained immunity. Regardless of mechanism, our data are consistent with those showing that the clonogenic potential of hematopoietic and stromal cells isolated from SCI patient bone marrow is impaired[21,22,78].

In conclusion, data in this report reveal that SCI-induced bone marrow failure is caused by cell-intrinsic and extrinsic mechanisms; impaired control of HSPC proliferation and sequestration is likely a cell-extrinsic phenomenon regulated in the niche while deficits in HSPC clonogenic potential are likely the result cell-intrinsic changes. Bone marrow failure develops soon after injury but has long-lasting adverse effects on the host HSPCs. For example, impaired bone marrow function may limit the development of an effective immune system, perhaps explaining the higher incidence of infectious morbidity and mortality in SCI patients. Also, SCI-induced bone marrow failure may preclude the use of bone marrow cells from SCI donors as a transplantation source. A similar limitation was recently described for bone marrow isolated from mice with experimental CNS autoimmune disease[79,80]. Together, these data highlight the bone marrow as a previously underappreciated therapeutic target for improving health outcomes and quality of life after SCI.

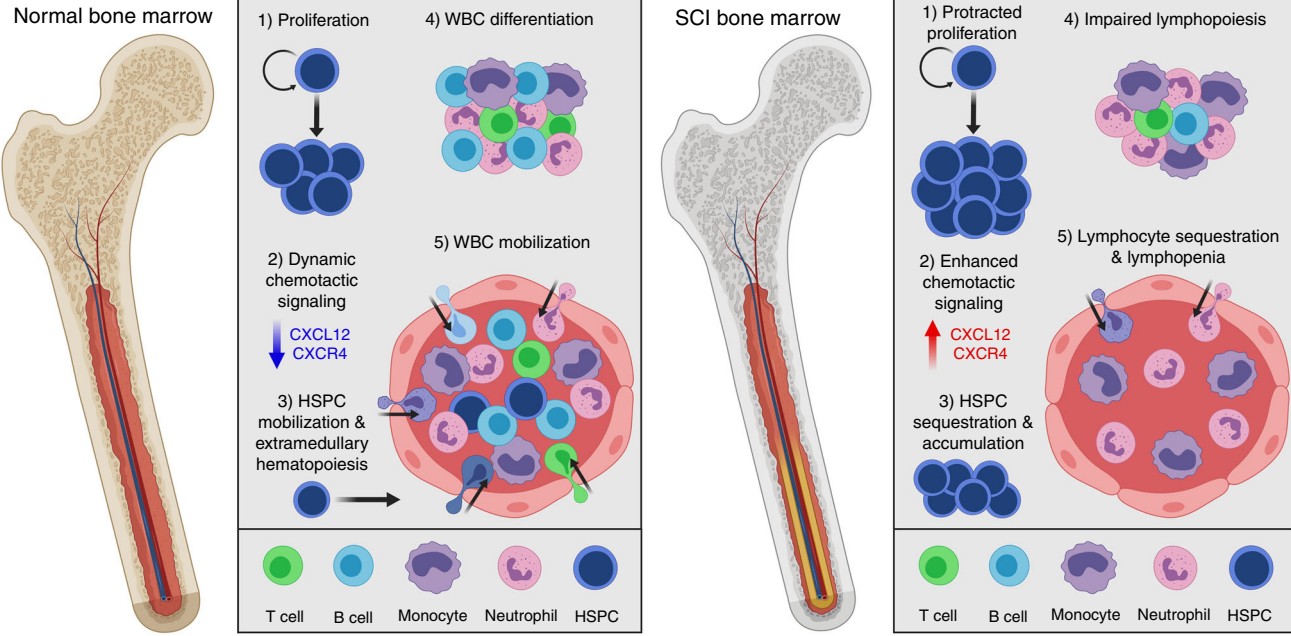

**Fig. 9 Summary of the effects of T3 transection SCI on bone marrow hematopoiesis.** Under homeostatic conditions, HSPCs undergo highly regulated and short-lived proliferation (1). When demand dictates (i.e. stress, trauma, infection), normal bone marrow downregulates CXCL12-CXCR4 signaling (2), allowing HSPC mobilization and extramedullary hematopoiesis (3). Homeostatic bone marrow also allows for the coordinated differentiation (4) and mobilization (5) of all leukocyte lineages, maintaining normal circulating blood leukocyte levels. T3 transection SCI, however, causes chronic bone marrow failure, including protracted proliferation (1), excessive CXCL12-CXCR4 chemotactic signaling (2), and impaired mobilization leading to the accumulation of HSPCs within the bone marrow and loss of extramedullary hematopoiesis. T3 transection SCI also causes lymphopenia by reducing lymphopoiesis (4) and sequestering mature lymphocytes in bone marrow (5).

## Methods

**Mice and housing**. The Institutional Animal Care and Use Committee of the Office of Responsible Research Practices at The Ohio State University approved all animal protocols for this study. All experiments were performed in accordance with the guidelines and regulations of The Ohio State University and outlined in the Guide for the Care and Use of Laboratory Animals from the National Institutes of Health. Female and male C57BL/6 mice (strain #000664; CD45.2) were purchased from The Jackson Laboratory (Bar Harbor, ME), female repTOP mitoIRE luciferase mice (Mito-luc) were purchased from Charles River Laboratories (Wilmington, MA), female B6.SJL-Ptprc$^a$ Pepc$^b$/BoyJ mice (C57BL/6-CD45.1; BoyJ) were bred in-house from adult breeding pairs originally purchased from The Jackson Laboratory (strain #002014), and male and female NOD.Cg-Prkdc$^{scid}$ Il2rg$^{tm1Wjl}$/SzJ mice (NSG mice) were bred in-house from adult breeding pairs purchased from The Jackson Laboratory (strain #005557). Animals were fed commercial food pellets and chlorinated reverse osmosis water ad libitum and housed (≤5/cage) in ventilated microisolator cages layered with corn cob bedding in a 12-h light–dark cycle at a constant temperature (20 ± 2 °C) and humidity (50 ± 20%). All mice were housed in a specific pathogen-free housing facility with routine testing of sentinel mice for specific pathogens. Generation of NSG mice with human immune systems (hNSG mice) was performed as follows[38–40]. Newborn NSG pups (24–72 h postnatal) received 1 Gy whole-body X-ray irradiation (RS 2000, Rad Source, Suwanee, GA), followed immediately by engrafting $1–5 \times 10^4$ human umbilical cord CD34$^+$ stem cells (Lonza Incorporated, Walkersville, MD or Stemcell Technologies, Vancouver, BC) via intrahepatic injection. Body temperature was maintained at 37 °C using a heating pad until pups were returned to their dams for normal maturation and weaning 21–24 days postnatal.

**SCI and animal care**. Adult C57BL/6 mice (10–16 weeks old) and hNSG mice (16–24 weeks old) were used for SCI experiments. Mice were subjected to a complete spinal cord transection injury at the third thoracic, ninth thoracic, or sixth lumbar spinal levels. Mice subjected to laminectomy (Lam or sham injury) only served as controls, and naïve mice were used as a reference for outcomes. Mice were anesthetized with ketamine (120 mg kg$^{-1}$, i.p.) and xylazine (10 mg kg$^{-1}$, i.p.) for all surgical procedures and provided prophylactic antibiotic treatment with gentamicin sulfate (5 mg kg$^{-1}$, s.q.). Aseptic conditions were maintained during all surgical procedures and mice were placed on a warming pad to regulate body temperature. Hair was shaved at the region of the thoracic spinal cord and skin was treated with a sequence of betadine, 70% ethanol, and betadine. A small midline incision was made to expose the vertebra and then a partial laminectomy was performed. The meninges were cut using micro-scissors and then spinal cord transected using micro-scissors and a sterile glass aspiration tube for suction of

fluid/blood, stabilization of the spinal cord during transection, and to confirm completion of injury. Muscle overlaying the injury site was sutured, followed by closure of the wound with sutures or staples. After surgery, mice were placed in cages on heating pads and monitored frequently until they recovered consciousness and were moving spontaneously. Mice were given fluids (1–2 mL 0.9% sterile saline) to maintain hydration and softened food to eat ad libitum as they recovered. Bladders were expressed at least two times daily to maintain urinary function, and urine underwent periodic pH testing to identify bladder infections. Gentocin antibiotic was subcutaneously administered once daily at 5 mg kg$^{-1}$ for 5 dpi. For contusion studies, identical surgical procedures were followed, with the exception of laminectomy level (T9). An Infinite Horizon Impactor (Precision Systems and Instrumentation, Lexington, KY) was used to generate a moderate 70 kdyne injury at either the third or ninth thoracic spinal levels.

**IVIS imaging**. Mito-luc mice were injected with 80 mg kg$^{-1}$ (i.p.) D-luciferin potassium salt (Cayman Chemical, Ann Arbor, MI), anesthetized with isoflurane (2.5–4% vaporized in oxygen, delivered at 1 L min$^{-1}$), and placed on a heated surface within the IVIS Lumina II system (Caliper Life Sciences, Hopkinton, MA) for image acquisition. Imaging of long bones in limbs required securing forelimbs at 90° from midline, and hindlimbs 45° from midline, with small pieces of translucent medical tape. Mice were kept on 1.5–2% isoflurane throughout the duration of imaging. Baseline levels of mitosis were determined prior to SCI, and then mice were imaged at 1, 3, 7, 14, 21, and 28 dpi. Data were analyzed with the Living Image® software (v.4.3.1; Caliper Life Sciences). Bioluminescence was measured by defining regions of interest (ROIs) of defined sizes around the sternum, left femur/tibia, and right femur/tibia. Total flux (photons s$^{-1}$) and the maximum radiance for each ROI was determined, and an average was calculated for both left and right femur/tibia ROIs. Data were plotted for each individual animal as either raw values (post-SCI study; Fig. 1) or percent of pre-LPS (chronic SCI plus LPS study; Fig. 7).

**Plerixafor treatment**. After SCI or sham surgery, mice were injected subcutaneously with 5 mg kg$^{-1}$ Plerixafor (AMD3100; Sigma-Aldrich) in 0.9% sterile saline. First dose was given 1-h post-injury, and then once a day until 3 dpi. Mice were terminally anesthetized with ketamine and xylazine 1 h after a final dose of AMD and tissues were collected as described below.

**Stimulating hematopoiesis with systemic endotoxin**. Mito-luc transgenic and wild-type mice underwent SCI or sham surgery as previously described. Approximately 6–8 weeks after injury mice were injected i.p. with 1 mg kg$^{-1}$ LPS (E. coli O55:B5, Sigma-Aldrich) in 0.9% sterile saline once per day for 3 days. Mito-

luc mice underwent IVIS imaging at 2, 4, 7, 9, 11, and 14 days post-LPS (first dose). Wild-type mice underwent submandibular bleeds prior to LPS, 1-h post-LPS, 4 days post-LPS, and 11 days post-LPS. Blood was collected into an EDTA-coated capillary tube (Sarstedt Inc.; Thermo Fisher Scientific, Waltham, MA).

**Tissue collection and processing**. Mice were terminally anesthetized with ketamine and xylazine for euthanasia and tissue collection. Blood was then collected via cardiac puncture and placed in blood collection tubes coated with EDTA. Blood was then treated with ammonium chloride-based red blood cell (RBC) lysis buffer and resuspended in Iscove's Modified Dulbecco's Medium (IMDM) with 2% fetal bovine serum (FBS) for downstream MethoCult assays or 0.1 M phosphate buffer saline (PBS) with 2% FBS (flow buffer) for flow cytometry. Spleens were rapidly isolated, weighed, and placed in Hank's Balanced Salt Solution (HBSS). Spleens were minced with sterile dissection scissors, smashed through a 40-μm sterile filter using the plunger of a 3-mL syringe, and rinsed with 10 mL of HBSS or IMDM. Mouse femurs and tibiae were removed, cleaned, and placed in a small volume of HBSS. Bone marrow cells were isolated by either flushing bones with 10 mL of HBSS or crushing in a mortar and pestle and washed with media. Cell counts were obtained by a standard hemocytometer (bone marrow and spleen), or with a Hemavet 950 fs multi-species hematology (blood; Drew Scientific, Miami Lakes, FL) system capable of analyzing whole blood with 5-part white blood cell differential, platelets, and RBCs.

**Immunolabeling and flow cytometry**. $2–10 \times 10^6$ bone marrow cells and splenocytes, or approximately 50 μL RBC-lysed blood, were allocated for flow cytometry analysis. All antibodies were used at a 1:100 dilution for staining purposes. BD Stemflow™ Mouse Hematopoietic Stem Cell Isolation Kit (BD Biosciences, cat #560492) was used to label lineage$^-$, c-Kit$^+$, Sca-1$^+$ HSPCs. Mouse antibody lineage cocktail (BD Biosciences; cat #558074) contained the following APC-conjugated antibodies: CD3 (145-2C11), CD11b (M1/70), CD45R/B220, TER-119, and Ly6G/C (RB6-8C5). Fc receptors were blocked for 15 min using rat anti-mouse CD16/32 antibody (BD Biosciences, cat #553142), followed by labeling with antibodies for 60 min. Dead cells were labeled with eFluor780 (eBioscience, cat #65-0865-14) approximately 30 min into antibody incubation. Labeled cells were fixed and permeabilized with BD Cytofix/Cytoperm™ solution (BD Biosciences, cat #554722) for 20 min. For cell cycle analysis, DNA was labeled with DAPI (BD Biosciences, cat #564907) in flow buffer with 0.1% Triton X-100 for 20 min after antibody labeling. For human HSPCs, antibodies for phosphorylated γH2AX (2F3; BioLegend cat #613414) were used to measure replication stress in c-Kit$^+$ HSPCs. For total CXCR4 receptor expression in LSK cells, mature bone marrow cells were depleted using Lineage Cell Depletion Kit and MACS system as per the manufacturer's protocol (Miltenyi Biotec, cat #130-090-858, Auburn, CA), followed by cell surface staining for LSK markers, fixation, permeabilization with BD Perm/Wash, and staining for CXCR4 (2B11; BD Biosciences, cat #562738). All incubations were performed at 4 °C, followed by a wash step using excess flow buffer, and centrifugation for 5 min at 4–10 °C. Antibodies for mouse lineage (BD Biosciences, cat #560492), human lineage (Invitrogen, cat #22-7778-72), human CD34 (581; BD Biosciences, cat #555824), and human CD38 (HIT2; BD Biosciences, cat #560677) were used to identify human HSPCs (Fig. 4d, Supplementary Fig. 1). Antibodies for CD3 (17A2; BD Biosciences, cat #564008), CD4 (RM4-5; BD Biosciences, cat #553052), CD24 (M1/69; BD Biosciences, cat #562563), CD43 (S7; BD Biosciences, cat #562865), CD45/B220 (RA3-6B2; BD Biosciences, cat #552772), IgM (II/41; BD Biosciences, cat #562032), and IgD (11-26c.2a; BD Biosciences, cat #562022) were used for analysis of bone marrow B and T cells (Fig. 5, Supplementary Fig. 1). Lineage cocktail (BioLegend, cat #133307) and antibodies for CD117 (2B8; BioLegend, cat #105827), Sca-1 (D7; BioLegend, cat #108142), CD48 (HM48.1; BioLegend, cat #103423), CD150 (TC15-12F12.2; BioLegend, cat #115916), CD135 (A2F10; BioLegend, cat #135305), and CD16/32 (93; BioLegend, .cat #101327) were used to identify LT-HSC/MPP1, ST-HSC, MPP2, MPP3, MPP4, GMP, and CMP/MEP subsets of HSPCs after LSK gating (Fig. 2). LSR II and LSR Fortessa flow cytometers (BD Biosciences) were used to analyze samples. Forward scatter and side scatter parameters were used to gate viable cell populations for phenotypic analysis (Supplementary Fig. 1). Positive and negative cell populations were selected based on staining with isotype control antibodies and fluorescent minus one control. Offline data analysis was completed with FlowJo v.10 software (Tree Star, Inc., Ashland, OR).

**MethoCult CFC assay**. The MethoCult™ GF M3434 ex vivo culture assay (Stemcell Technologies, Vancouver, BC) consists of a methocellulose media containing cytokines and growth factors to support the development of cell colonies from single HSPCs. Bone marrow cells were plated at concentrations ranging from $7.5 \times 10^3$, splenocytes at $2 \times 10^5$, and RBC-lysed blood at 150 μL in a total of 1 mL of MethoCult media. Samples were plated in meniscus-free six-well SmartDish™ (Stemcell Technologies) and placed in an incubator at 37 °C and 5% $CO_2$. Approximately 10–12 days after plating, colonies were quantified by standard inverted light microscopy using a StemGrid™ counting underlay (Stemcell Technologies). Colonies were identified as blast forming unit-erythrocyte, granulocyte, monocyte, granulocyte/monocyte, or granulocyte/erythrocyte/monocyte/megakaryocyte based on colony composition and cell morphology. MethoCult™

H4034 Optimum was used for quantification of human CFCs isolated and purified from humanized mice.

**CRU assay**. The CRU assay was used to assess repopulation potential of whole-bone marrow cells isolated from naïve, lam, and SCI mice. Bone marrow cells ($5 \times 10^6$) from donor C57BL/6 mice (3 dpi) expressing leukocyte antigen CD45.2 were isolated and mixed with equal numbers of rescue bone marrow from BoyJ mice expressing leukocyte antigen CD45.1. This mixture was injected via lateral tail vein into lethally irradiated BoyJ recipient mice (4–6 weeks old; total 9 Gy split into two doses of 4.5 Gy approximately 24 h apart; gamma irradiation from Cesium-137). Donor chimerism was assessed every 4 weeks after engraftment via flow cytometry for CD45.1 (BoyJ) and CD45.2 (donor) peripheral blood leukocytes and expressed as a percent of total leukocytes: CD45.2/(CD45.1+CD45.2). Bone marrow, spleen, and blood were collected 19 weeks after engraftment for donor chimerism and blood phenotyping using CD4, CD8, CD45/B220, and CD11b antibodies. Secondary engraftment was performed after isolating donor CD45.2 bone marrow cells ($5 \times 10^6$) from primary recipients using magnetic bead antibodies and columns. Isolated cells were mixed with equal numbers of newly isolated CD45.1 BoyJ bone marrow and engrafted into new BoyJ recipients.

**RT-qPCR of cytokine and chemokine mRNA**. Bone marrow was isolated and pooled from tibia and femurs of laminectomy and T3Tx mice ($n = 3$/group). RNA was isolated using TRIzol as per the manufacturer's instructions. Genomic DNA was eliminated using 1 μg μL$^{-1}$ DNase I (Invitrogen). One microgram DNase-treated RNA was primed with random hexamers (1 μM; Applied Biosystems) and reverse transcribed into cDNA using SuperScript II reverse transcription (Applied Biosystems) in a 20-μL reaction. RNase-free sterile water was used to dilute 1 μg cDNA 1:10, loaded onto a 96-well plate, and then primers were loaded (Supplementary Table 1). SYBR Green Master Mix (Applied Biosystems) was used to detect amplified cDNA. Melting point curves assessed the quality of each reaction. Samples were run in triplicates, gene expression was normalized to s18 control samples using the delta-delta CT method, and SCI data were expressed as fold increase of sham-injured values.

**Quantification of CXCL12 in bone marrow extracellular fluid**. Bone marrow was isolated from two tibia and two femurs per mouse using 1 mL ice cold PBS. After light trituration, cells and debris were separated from extracellular fluid by centrifugation for 5 min at $400 \times g$. Fluid was aliquoted and frozen in liquid nitrogen. A Mouse CXCL12/SDF-1α Quantikine ELISA kit (R&D Systems) was used as per the manufacturer's protocol. Bone marrow extracellular fluid preps were measured undiluted. Plates were read at 450 nm wavelength on a SpectraMAX190 and analyzed using the SoftMax Pro software (Molecular Devices, San Jose, CA).

**Statistics and reproducibility**. All data are represented as mean ± standard error of the mean (SEM), with individual data points representing independent biological replicates. Group sizes were determined by analyzing preliminary and published data; using G*Power (v3.1), an $n = 4$ was found to be sufficient to detect a 1.25-fold change with a coefficient of variation of 20% and >80% power for flow cytometry and MethoCult assays. To compensate for unexpected morbidity/mortality, an $n = 6$/group was determined. Except for the CRU assay, reference data from naïve C57BL/6J (10–16 weeks old over) were collected during development and testing of endpoint assays and are included as dotted lines on graphs. Data were excluded from analysis only if identified as a statistical outlier by Grubbs' test with proof of deviation from standard recovery after SCI (i.e. bladder infection, post-operative autophagia, etc.). A single female T3 transection animal met these criteria and was removed from Figs. 3 and 4 due to a documented bladder infection and blood CFCs > 800 mL$^{-1}$. To compare groups of SCI (T3Tx), when only one time point involved, we used two-sample $t$-tests (with Welch's $t$-test for unequal variances). For the experiment comparing the effect among sham, low-thoracic, and high-thoracic injuries (T3Tx vs T9Tx; Fig. 4a, b), one-way ANOVA was performed followed by Tukey post hoc comparisons. For the longitudinal measures (e.g. proliferation, BM engraftment; Figs. 1, 7, and 8), mixed effect or two-way ANOVA with repeated measures was conducted followed by Bonferroni multiple comparisons. All statistical tests were two-tailed. Exact $p$ values denoted on graphs; *$p < 0.05$ (Fig. 5b). Samples were blinded either during processing or prior to analysis by a separate experimenter not involved in the analysis. All attempts at replication were successful: BM (Fig. 1a–c) and HSPC (Fig. 1d) proliferation data are from two independent experimental replications. Accumulation of HSPCs was demonstrated in Fig. 1e–h, and then verified in an independent experimental replicate in Fig. 2. Accumulation of cells in tibia (Fig. 1h) has been confirmed with an independent experimental replication not included in the manuscript. Impaired HSPC mobilization after SCI was demonstrated in independent experimental replications using female and then male wild-type mice (Fig. 3) and further replicated in other SCI models (Fig. 4). Impaired HSPC mobilization was then confirmed with mice with human HSPCs (Fig. 4c, d). In total, impaired HSPC mobilization after SCI was confirmed in at least six independent experimental replications. Data demonstrating lymphopenia after T3 SCI (Fig. 5a, b) were from two independent experimental replications. Data demonstrating enhanced CXCL12-CXCR4 levels (Fig. 6b, c) are from two independent

experimental replications. Data demonstrating that Plerixafor liberates HSPCs from bone marrow after SCI (Fig. 6d–k) were independently verified in two additional experiments not included in the manuscript. Data were analyzed using GraphPad Prism software v5.0 (GraphPad Software Inc., San Diego, CA). Illustrations were created with BioRender scientific illustration software paid subscription (biorender.com). Figures were generated in Adobe Photoshop CS5 v12 (Adobe Systems Inc., San Jose, CA).

**Reporting summary**. Further information on research design is available in the Nature Research Reporting Summary linked to this article.

## Data availability

Source data are provided with this paper as a Source Data file, including all data for Figs. 1–8 (and Supplementary Fig. 2) which support the findings of the current study. All other information is available within the manuscript, supplementary information file, or by reasonable request from the corresponding author. Source data are provided with this paper.

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

## Acknowledgements

We would like to thank Zhen Guan, Feng Qin Yin, Vishal Kalbavi, and Kristina Kigerl for their technical assistance. We would also like to thank all the other members of the McTigue, Popovich, and Dorrance labs for their constructive feedback and support. Flow cytometry data in this report were collected using instruments and services at the Analytical Cytometry Core, The Ohio State University, supported by NCI P30CA016058. IVIS imaging was performed using instruments and services of the Small Animal Imaging Core lab, The Ohio State University. Automated CBCs were analyzed using the instruments and services of the Comparative Pathology and Mouse Phenotyping Shared Resource Core, The Ohio State University, supported by NCI P30CA016058. Experiments were performed with assistance from The Ohio State University Neuroscience Department Surgical Core supported by NINDS P30NS104177. This study was supported in part by a NIH Fellowship F31NS100303 (to R.S.C.), NIH Grant R35NS111582 (to P.G.P.), Craig H. Neilsen Foundation Pilot Grant 340884 (to P.G.P.), Craig H. Neilsen Postdoctoral Fellowship 457267 (to F.H.B.), Ohio State University President's Postdoctoral Scholar Award (to K.A.M.), and the OSU Ray W. Poppleton Endowment (to P.G.P.). Funding agencies were not involved in the study design, collection or interpretation of data, or preparation of the manuscript.

## Author contributions

R.S.C., A.M.D., and P.G.P designed the experiments; R.S.C. performed all experiments; J.M.M., F.H.B., K.A.M., J.C.E.H., and R.R.J. helped with experiments for Figs. 1–7; M.K., M.H.B., and A.M.D. helped perform CRU assay experiments for Fig. 8; R.S.C performed data analysis and generated figures; X.M.M. provided statistical analysis support; R.S.C., A.M.D., and P.G.P. wrote the manuscript; all authors contributed intellectually to the study, aided in manuscript preparation and revisions, and approved the final manuscript.

## Competing interests

The authors declare no competing interests.
