## [Peer Review File · Nature Communications]

Reviewers' comments:

Reviewer #1 (Spinal cord injury, neuro-immune crosstalk) (Remarks to the Author):

The authors report in this study that SCI results in the retention of hematopoietic stem and progenitor populations along with several mature immune cells within the bone marrow. The work is a further extension to the authors' large pool of work on the immunological changes following SCI, and with some additional experiments can offer key mechanistic insight into the phenomenon of chronic immune dysfunction following SCI. That said, I have several questions for the authors.

Major comments:

- As the authors have previously shown level-dependent differences in the spleen after SCI, they should clarify what model they are using in the SCIs following Fig 3 (it is assumed to be T3Tx).

Loss of supraspinal input and its impact on HSPC retention in the bone marrow

- Retrograde tracing demonstrates innervation of the bone marrow from the T8-L1 segments of the spine [1]. Specifically, if the retention / egress of HSPCs and mature immune cells is indeed dependent on sympathetic control, the authors cannot rule out a lack of level-dependent response unless a transection was done below the L1 segment – which would be a key mechanistic experiment. It is also important to note that a recent publication [2] showed that contusion injuries—a clinically-relevant model of SCI—may elicit a significantly different level-dependent adrenergic and catecholamine response relative to transection injuries. That is, in the case of incomplete injuries where sympathetic tone is not entirely lost, a T3 contusion may not be expected to show HSPC retention. As such, an experiment showing T3 contusion and its circulating HSPCs is warranted.

HSPC retention and its impact on susceptibility to pathogens

- In the current study, it was clear through Fig 3 that no level-dependence was shown in the transection model. The discussion did not clearly address the potential impact of HSPCs retention in the bone marrow when it occurs concurrently to splenic atrophy and splenocyte death. Does it exacerbate the ability of the animal to fight off infections? How is the clearance of infection in T3Tx vs. T9Tx with and without AMD?

Aberrant chemotactic signaling in the bone marrow and its impact on circulating chemotaxins

- Does the change in chemotactic signaling within the bone marrow reflect a change in the chemotactic signaling in the blood and spleen? That is, does an increase in chemotaxins associated with leukocyte sequestration in the bone marrow mirror a reduction in the chemotactic signaling in the circulation. Since chemotaxis is dependent on gradients in concentration, it is important to address this prior to concluding that a chemotactic mechanism is responsible for HSPC retention as AMD was administered systemically.

References

1. Nance, D. M., & Sanders, V. M. (2007). Autonomic innervation and regulation of the immune system (1987-2007). *Brain, Behavior, and Immunity*, 21(6), 736–745. <http://doi.org/10.1016/j.bbi.2007.03.008>
2. Hong, J., Chang, A., Liu, Y., Wang, J., & Fehlings, M. G. (2019). Incomplete Spinal Cord Injury

Reverses the Level-Dependence of Spinal Cord Injury Immune Deficiency Syndrome. *International Journal of Molecular Sciences*, 20(15), 3762. <http://doi.org/10.3390/ijms20153762>

Reviewer #2 (Stress-induced hematopoiesis)(Remarks to the Author):

This is a very interesting paper, addressing an important topic, the response of the bone marrow to spinal cord injury. The authors show that spinal cord injury caused enhanced proliferation of HSPCs and their sequestration in the bone marrow, as their mobilization was inhibited due to alterations in the CXCL12/CXCR4 axis. Consequently AMD3100 reversed the spinal cord injury effects on the bone marrow. Importantly, competitive repopulation assays showed that the functionality of HSCs was diminished upon spinal cord injury. The topic is very important. The manuscript is novel, contains potentially important data and is well written. However, important points need to be addressed:

1) Fig. 1: The analysis of cell populations in the bone marrow after SCI is rather rudimentary. The authors must provide a more thorough flow cytometry analysis at the different time points. They need to add CD150, CD48 staining in their LSK staining to analyze LT-HSC (especially given the data shown in the transplantation experiments in Fig. 7 suggesting that LT-HSCs may be dysfunctional) and MPPs. CLPs, CMPs, GMPs and MEPs should also be analyzed.

In addition analysis of proliferation in the aforementioned HSPC populations would be helpful. Is there also a change in subpopulations of HSPCs? For instance, are myeloid-biased CD41+ LT-HSCs altered? What happens to MPP2, MPP3, MPP4 subpopulations (especially as MPP3 are myeloid-biased and MPP4 lymphoid biased)?

This detailed analysis is critical to better understand the effects of SCI on the bone marrow.

2) Do HSPCs experience replication stress? This can be addressed by flow cytometry for gamma-H2Ax in LT-HSCs, MPPs, LSKs.

3) Fig. 5: What happens to the levels of other cytokines and growth factors in the bone marrow, such as G-CSF, GM-CSF, IL1b, TNF, IL6, IFN? Those may regulate HSPC proliferation!

4) Transplantation experiments in Fig. 7. The authors do not show the lineage output from the donor cells. They only show engraftment. What happens to myeloid cells and lymphocytes in the blood? Is there a bias? What happens to HSPCs, progenitors and mature myeloid cells and lymphocytes in the bone marrow? Is there a bias?

5) The long-term and transmissible effects, as shown in Fig. 6 and Fig. 7 can also be interpreted as a form of negative / detrimental trained immunity. In this context, different groups have shown long-term effects of trained immunity on HSPCs (e.g. <https://www.ncbi.nlm.nih.gov/pubmed/29328911> <https://www.ncbi.nlm.nih.gov/pubmed/29328910> <https://www.ncbi.nlm.nih.gov/pubmed/31213716> <https://www.ncbi.nlm.nih.gov/pubmed/29328912>). In my opinion, the authors should discuss their findings also in light of trained immunity and cite those 4 papers.

Response to Reviewer #1:

Comment 1: As the authors have previously shown level-dependent differences in the spleen after SCI, they should clarify what model they are using in the SCIs following Fig 3 (it is assumed to be T3Tx).

- Response: The effects of the nature/type (e.g., complete transection, contusion) and level (T3, T9, L6/S1) of spinal cord injury (SCI) on hematopoiesis are now shown in revised Fig. 4. All data following Fig. 4 were generated using a high-level (T3) SCI. These facts have been clarified in the revised text for Figs. 5-8.

Comment 2a) Retrograde tracing demonstrates innervation of the bone marrow from the T8-L1 segments of the spine [1]. Specifically, if the retention / egress of HSPCs and mature immune cells is indeed dependent on sympathetic control, the authors cannot rule out a lack of level-dependent response unless a transection was done below the L1 segment – which would be a key mechanistic experiment.

- Response: In comparing T3 and T9 injuries (original Fig. 3 data), our intent was to determine whether SCI-dependent changes in HSPC retention after high-level (T3) SCI were the result of altered sympathetic reflexes. Indeed, a complete T3 transection model of SCI eliminates all descending brain/brainstem input to the sympathetic preganglionic neurons (SPNs) in the spinal cord, including those that innervate femoral bone marrow between T8-L1. Conversely, after complete (transection) lesions at T9, descending input would be preserved to SPNs from at least one spinal level (T8). Thus, some sympathetic reflex control over bone marrow was expected after T9 SCI. Even more was expected after a T9 contusion injury, a model in which some descending brain/brainstem axons survive at the site of injury and may be able to modulate SPNs below the level of injury. However, to definitely test this hypothesis, as recommended by Reviewer 1, we placed a complete spinal transection injury at L6/S1 spinal segment (L1 vertebral body). This injury should effectively preserve all descending control of sympathetic innervation to the bone marrow (and other organs throughout the body). However, like all other injuries that we tested, SCI prevented HSPC mobilization (compared to mice receiving only a sham/ laminectomy surgery at the same spinal level; *new data included in revised Fig. 4*). See additional commentary below in response to Comment 2b.

Comment 2b) It is also important to note that a recent publication showed that contusion injuries—a clinically-relevant model of SCI—may elicit a significantly different level-dependent adrenergic and catecholamine response relative to transection injuries. That is, in the case of incomplete injuries where sympathetic tone is not entirely lost, a T3 contusion may not be expected to show HSPC retention. As such, an experiment showing T3 contusion and its circulating HSPCs is warranted.

- Response: As suggested, we delivered controlled spinal contusion injuries at the T3 spinal cord level. Although this model of T3 contusion injury has not been fully characterized, some residual hindlimb function persisted in each mouse (open-field locomotor scoring (BMS scores) revealed consistent BMS scores of ~2.5 indicating extensive ankle movement and/or plantar placement of the foot during stepping), indicating that the contusion lesions were indeed, incomplete (i.e., sparing some axons at the injury epicenter). Like all other SCI mice, regardless of whether the injury was anatomically complete/incomplete at high thoracic or lumbar/sacral spinal levels, HSPCs were retained in bone marrow (*see revised data in Fig. 4B*). When all data are considered together, including new data obtained with the L6/S1 SCI model (see above), we can conclude that the loss of inhibitory signals to spinal sympathetic preganglionic neurons (SPNs) from brain/brainstem is unlikely to be a primary trigger for the enhanced HSPC retention in the bone marrow after SCI. However, injury-induced changes in spinal sympathetic reflex control of bone marrow may still contribute to bone marrow failure after SCI. The revised Discussion includes a detailed explanation for how this can occur along with an extended discussion of how altered sympathetic-neuroendocrine reflexes may contribute to HSPC retention in bone marrow. Appropriate references have been added.

Comment 3) In the current study, it was clear through Fig 3 that no level-dependence was shown in the transection model. The discussion did not clearly address the potential impact of HSPCs retention in the bone

marrow when it occurs concurrently to splenic atrophy and splenocyte death. Does it exacerbate the ability of the animal to fight off infections? How is the clearance of infection in T3Tx vs. T9Tx with and without AMD?

- Response: In our original Discussion we did speculate on the potential immunological significance of sequestration of HSPCs and mature leukocytes in bone marrow after SCI. We did not address the relationship between HPSCs sequestration in bone marrow and splenic atrophy/splenic leukocyte death. This was an omission by choice because we have not yet tested whether AMD, which effectively mobilizes sequestered HSPCs and leukocytes from bone marrow and restores extramedullary hematopoiesis, can also restore immune function. This is an important question and we have started experiments that I hope, with time, will answer some or all of the above questions. Specifically, a new postdoc in the lab is working on a project that will seek to understand why SCI impairs immune surveillance in the lung. Her pilot data indicate that after T3 SCI, fewer immune cells (both resident lung immune cells and leukocytes recruited from the blood) are present in the lung (data not shown) and that AMD, the same CXCR4 antagonist used to overcome HSPC and leukocyte sequestration in bone marrow, can increase total leukocyte numbers in the lung (see *inserted data at right showing the number of leukocytes in the lungs after T3 transection SCI in mice treated with vehicle or AMD3100 1x/d for 3 days; these data were not added to revised text*). Additional work is needed to fully characterize the immune response in the lung after SCI, and whether impaired immune surveillance in lung after SCI is caused by a reduction in the number and/or function of resident or recruited leukocytes. Also, even though AMD releases cells from bone-marrow, we do not know for how long these effects persist or even if the immune cells that are released are functional. The reviewer's questions are clearly relevant and are consistent with current work in the lab but answers to these questions are not yet available and are beyond the scope of the current manuscript.

Comment 4) Does the change in chemotactic signaling within the bone marrow reflect a change in the chemotactic signaling in the blood and spleen? That is, does an increase in chemotaxins associated with leukocyte sequestration in the bone marrow mirror a reduction in the chemotactic signaling in the circulation. Since chemotaxis is dependent on gradients in concentration, it is important to address this prior to concluding that a chemotactic mechanism is responsible for HSPC retention as AMD was administered systemically.

- Response: It never occurred to us to measure CXCL12 in blood or spleen because that is not what causes HSPCs and leukocytes to mobilize from bone marrow. Instead, it is well-known that the retention and release of HSPCs and leukocytes from bone marrow are regulated by CXCL12 gradients in the bone marrow. To our knowledge there are no published data showing that levels of CXCL12 increase under physiological or pathological conditions to regulate HSPC or white blood cell egress from the bone marrow. However, published data do show that when injected systemically, AMD3100 acts in the bone marrow to inhibit CXCL12-CXCR4 signaling and promote the egress of HSPCs and leukocytes into circulation⁴. These facts have been emphasized in the revised Discussion and additional references were added.

4. Liu, Q., Li, Z., Gao, J.L., Wan, W., Ganesan, S., McDermott, D.H., and Murphy, P.M. (2015). CXCR4 antagonist AMD3100 redistributes leukocytes from primary immune organs to secondary immune organs, lung, and blood in mice. *Eur. J. Immunol.* 45, 1855–1867.

Response to Reviewer #2:

Comment 1) Fig. 1: The analysis of cell populations in the bone marrow after SCI is rather rudimentary. The authors must provide a more thorough flow cytometry analysis at the different time points. They need to add CD150, CD48 staining in their LSK staining to analyze LT-HSC (especially given the data shown in the transplantation experiments in Fig. 7 suggesting that LT-HSCs may be dysfunctional) and MPPs. CLPs, CMPs, GMPs and MEPs should also be analyzed. In addition analysis of proliferation in the aforementioned HSPC populations would be helpful. Is there also a change in subpopulations of HSPCs? For instance, are myeloid-biased CD41+ LT-HSCs altered? What happens to MPP2, MPP3, MPP4 subpopulations (especially as MPP3 are myeloid-biased and MPP4 lymphoid biased)? This detailed analysis is critical to better understand the effects of SCI on the bone marrow.

- Response: As recommended, we expanded our phenotypic analyses of HSPCs. New data in revised Fig. 2 confirm our previous data and show that SCI causes prolonged accumulation of HSPCs and total number of bone marrow cells (at least until 28dpi). These new data also show an accumulation of the earliest stem cells (CD150+/48-/135- LSKs) and various differentiated progenitor cell subsets. While all LSK subsets show increases in total numbers, we also examined the composition (i.e., proportion) of LSK cells by subset. A few minor differences were noted. Particularly, we found slightly increased proportion of MPP4s, and fewer ST-HSCs and MPP3s, make up the LSK compartment in chronic SCI mice. The results section was revised accordingly.

Comment 2) Do HSPCs experience replication stress? This can be addressed by flow cytometry for gamma-H2Ax in LT-HSCs, MPPs, LSKs.

- Response: Whether excessive proliferation of HSPCs leads to accumulation of replication stress (DNA breaks, etc.) is an interesting question. We performed intra-nuclear staining for serine 139 phosphorylation of gamma-H2Ax (indicating double stranded DNA breaks) in c-Kit+ cells isolated from bone marrow of sham-injured and SCI mice at 28d post-surgery/injury. The results indicate that HSPCs do not experience replication stress, even at late post-injury stages. These data have been added to revised Fig. 2.

Comment 3) Fig. 5: What happens to the levels of other cytokines and growth factors in the bone marrow, such as G-CSF, GM-CSF, IL1b, TNF, IL6, IFN? Those may regulate HSPC proliferation!

- Response: New cytokine/chemokine PCR data are included in the revised Results (see new Fig. 6A). Also, based on this reviewer's comment, we have included in the revised text more detailed rationale to explain why, despite there being other mechanisms to explore (based on the PCR data), we chose to further characterize then validate only CXCL12/CXCR4. The revised Results section now includes two revised paragraphs.

Comment 4) Transplantation experiments in Fig. 7. The authors do not show the lineage output from the donor cells. They only show engraftment. What happens to myeloid cells and lymphocytes in the blood? Is there a bias? What happens to HSPCs, progenitors and mature myeloid cells and lymphocytes in the bone marrow? Is there a bias?

- Response: We did analyze lineage output at the end of the primary engraftment experiment. Specifically, we quantified donor lymphocytes (CD4+ and CD8+ T cells and B220+ B cells) and CD11b+ myeloid cells in blood at 19 weeks post-engraftment (see Suppl Fig. 2E). Based on those data, we can conclude that SCI does not bias the development of engrafted cells towards a specific cell lineage. Text in the results has been updated to specifically reference these supplemental data. Also, consistent with those data, new HSPC phenotyping data (see revised Fig. 2) indicate that SCI does not impose a significant bias in the bone marrow niche – there is a SCI-dependent increase in all HSCs and multipotent progenitor cells.

Comment 5) The long-term and transmissible effects, as shown in Fig. 6 and Fig. 7 can also be interpreted as a form of negative / detrimental trained immunity. In this context, different groups have shown long-term effects of trained immunity on HSPCs (e.g. <https://www.ncbi.nlm.nih.gov/pubmed/29328911> <https://www.ncbi.nlm.nih.gov/pubmed/29328910> <https://www.ncbi.nlm.nih.gov/pubmed/31213716> <https://www.ncbi.nlm.nih.gov/pubmed/29328912>). In my opinion, the authors should discuss their findings also in light of trained immunity and cite those 4 papers.

- We agree that the long-term and transmissible effects of SCI on HSPCs that we describe in (revised Figs. 7&8) could be explained by some form of epigenetic modification of HSPCs, creating a form of negative/detrimental trained immunity. We have revised the discussion to include this possibility and cited the appropriate references.

REVIEWERS' COMMENTS:

Reviewer #1 (Remarks to the Author):

The authors conducted several additional mechanistic experiments (lumbar transection and high thoracic contusion) to demonstrate that bone marrow failure is a level-independent phenomenon after SCI, and that loss of input from SPNs is not a likely trigger for retention of HSCs. I'm sufficiently convinced that SCI-induced alterations in the neuroendocrine response may be a critical determinant of chronic bone marrow failure. The authors also described the questions this raises about the level-dependent immune response observed in their previous work on transection models and presents some unpublished data to demonstrate some early investigations into these questions.

Reviewer #2 (Remarks to the Author):

The authors have addressed all my comments. This is an excellent paper.

Reviewer #1 (Remarks to the Author):

The authors conducted several additional mechanistic experiments (lumbar transection and high thoracic contusion) to demonstrate that bone marrow failure is a level-independent phenomenon after SCI, and that loss of input from SPNs is not a likely trigger for retention of HSCs. I'm sufficiently convinced that SCI-induced alterations in the neuroendocrine response may be a critical determinant of chronic bone marrow failure. The authors also described the questions this raises about the level-dependent immune response observed in their previous work on transection models and presents some unpublished data to demonstrate some early investigations into these questions.

Reviewer #2 (Remarks to the Author):

The authors have addressed all my comments. This is an excellent paper.